# Pathogenicity of *Beauveria bassiana* PfBb and Immune Responses of a Non-Target Host, *Spodoptera frugiperda* (Lepidoptera: Noctuidae)

**DOI:** 10.3390/insects13100914

**Published:** 2022-10-08

**Authors:** Yi-Ping Gao, Mei Luo, Xiao-Yun Wang, Xiong Zhao He, Wen Lu, Xia-Lin Zheng

**Affiliations:** 1Guangxi Key Laboratory of Agric-Environment and Agric-Products Safety, College of Agriculture, Guangxi University, Nanning 530004, China; 2School of Agriculture and Environment, Massey University, Private Bag 11222, Palmerston North 4410, New Zealand

**Keywords:** *Spodoptera frugiperda*, *Beauveria bassiana*, virulence, non-target host, enzyme activity

## Abstract

**Simple Summary:**

In this study, we tested the pathogenicity of *Beauveria bassiana* PfBb on the important agricultural pest *Spodoptera frugiperda* (Lepidoptera: Noctuidae) by determining the relative activities of protective enzymes and detoxifying enzymes in different larval instars. Our results show that the *B. bassiana* PfBb strain could infect all six larval instars of *S. frugiperda*, and its virulence to *S. frugiperda* larvae gradually increased with an increase in spore concentration. Furthermore, the activities of protective enzymes (i.e., catalase, peroxidase, and superoxide dismutase) and detoxifying enzymes (i.e., glutathione S-transferases, carboxylesterase, and cytochrome P450) in *S. frugiperda* larvae of the first three instars infected with *B. bassiana* PfBb changed significantly with infection time, but such variations were not obvious in the fifth and sixth instars. These findings laid the foundation for further research on the mechanism by which *B. bassiana* controls *S. frugiperda*.

**Abstract:**

Exploring the pathogenicity of a new fungus strain to non-target host pests can provide essential information on a large scale for potential application in pest control. In this study, we tested the pathogenicity of *Beauveria bassiana* PfBb on the important agricultural pest *Spodoptera frugiperda* (Lepidoptera: Noctuidae) by determining the relative activities of protective enzymes and detoxifying enzymes in different larval instars. Our results show that the *B. bassiana* PfBb strain could infect all six larval instars of *S. frugiperda*, and its virulence to *S. frugiperda* larvae gradually increased with an increase in spore concentration. Seven days after inoculation, the LC_50_ of *B. bassiana* PfBb was 7.7 × 10^5^, 5.5 × 10^6^, 2.2 × 10^7^, 3.1 × 10^8^, 9.6 × 10^8^, and 2.5 × 10^11^ spores/mL for first to sixth instars of *S. frugiperda*, respectively, and the LC_50_ and LC_90_ of *B. bassiana* PfBb for each *S. frugiperda* instar decreased with infection time, indicating a significant dose effect. Furthermore, the virulence of *B. bassiana* PfBb to *S. frugiperda* larvae gradually decreased with an increase in larval instar. The activities of protective enzymes (i.e., catalase, peroxidase, and superoxide dismutase) and detoxifying enzymes (i.e., glutathione *S*-transferases, carboxylesterase, and cytochrome P450) in *S. frugiperda* larvae of the first three instars infected with *B. bassiana* PfBb changed significantly with infection time, but such variations were not obvious in the fifth and sixth instars. Additionally, after being infected with *B. bassiana* PfBb, the activities of protective enzymes and detoxification enzymes in *S. frugiperda* larvae usually lasted from 12 to 48 h, which was significantly longer than the control. These results indicate that the pathogenicity of *B. bassiana* PfBb on the non-target host *S. frugiperda* was significant but depended on the instar stage. Therefore, the findings of this study suggest that *B. bassiana* PfBb can be used as a bio-insecticide to control young larvae of *S. frugiperda* in an integrated pest management program.

## 1. Introduction

Insect epidemics caused by entomopathogenic fungi can naturally suppress pest populations, reduce pesticide use, and maintain ecological balance. A wide range of entomopathogenic fungi, particularly those belonging to the *Beauveria*, *Isaria*, *Lecanicillium,* and *Metarhizium* genera, have been widely studied [1,2,3,4]. However, an improved understanding of the pathogenicity of entomogenous fungi is a prerequisite for evaluating pest control efficiency. To date, some entomopathogenic fungi have been commercialized and widely used for pest management in greenhouses and fields, especially *Beauveria bassiana* [5,6,7]. Many studies have shown that *B. bassiana* has significant lethal effects on a variety of insect pests, such as *Ostrinia furnacalis* [8], *Cylas formicarius* [9], *Bemisia tabaci* [10], *Lycorma delicatula* [5], *Frankliniella occidentalis* [6], and *Leptinotarsa decemlineata* [4]. Although *B. bassiana* has strong host specificity, the virulence of different strains to non-target host pests varies greatly [11,12,13]. Therefore, clarifying the virulence of a new *B. bassiana* strain to non-target host pests can provide essential information for potential application in pest control on a large scale.

Insects are usually exposed to xenobiotics in the environment, in which their successful adaptation to these environmental risks requires an efficient system to detoxify and eliminate these substances from their bodies. For example, *B. bassiana* affects the normal physiological and metabolic activities in infected insects [14]; it causes damage to the free amino acids in the hemolymph, and interferes with a variety of important metabolic enzymes, such as glutathione S-transferases (GST) [15], carboxylesterase (CarE) [16], and cytochrome P450 (CYP450) [17], which play an important role in the detoxification of penetrated xenobiotics [18,19]. In addition, the protective enzyme systems in insects, mainly including catalase (CAT), peroxidase (POD), and superoxide dismutase (SOD), could reduce oxidative damage by degrading H_2_O_2_ [20]. The toxic reactive oxygen species (ROS) are generated and accumulated by the innate immune cells of organisms when insects are under stressful conditions, such as the applications of chemical pesticides and entomopathogenic fungi [21,22,23]. In many studies, to maintain normal cellular function and reduce oxidative damage, different types of ROS are scavenged by protective enzymes [24,25]. Therefore, understanding the activity of detoxifying enzymes and protective enzyme systems is helpful for exploring the resistance mechanism of insects during fungal infection, which is of great significance for pest control.

The fall armyworm (FAW), *Spodoptera frugiperda* (J.E. Smith) (Lepidoptera: Noctuidae), is a notorious migratory pest originating from the Americas [26]. It invaded Africa in 2016 [27], and then Southeast Asian countries such as India [28], Myanmar, and Vietnam in 2018 [29]. *Spodoptera frugiperda* was first recorded in Yunnan, China, in 2019 [30]. It has become a major agricultural pest across borders and continents due to its wide host range, high reproductive capacity, and fast dispersal ability [31]. A variety of natural enemies attack *Spodoptera frugiperda*, but they have rarely been used in the biological control of this pest in China [32]. Thus far, the control of *S. frugiperda* has mainly relied on chemical insecticides [33,34]. However, the long-term application of insecticides could result in the development of insecticide resistance and damage to the environment. Alternatively, a large number of studies have shown that *B. bassiana* is pathogenic to *S. frugiperda* [35,36], and that *B. bassiana* could be compatible with natural enemies to improve the control effect [37]. Therefore, entomopathogenic fungi could be prime substitutes for insecticides [38,39]. However, most entomopathogenic isolates show high mortality only to young larvae but low pathogenicity to old larvae [40]. The potential causes are still unknown.

The entomopathogenic fungus, *B. bassiana* PfBb strain, was isolated from the naturally infected dead larvae of the diurnal red moth, *Phauda flammans,* in Du’an County (108°8′ E, 23°52′ N), Hechi City, Guangxi Zhuang Autonomous Region, China. Our previous studies confirmed that the *B. bassiana* PfBb strain has strong pathogenicity to *P. flammans* larvae compared with other entomopathogenic fungi [41,42]. For example, the mortality of *P. flammans* larvae observed after treatment with *B. bassiana* PfBb (97.50 ± 2.89%) is significantly higher than those infected with *Metarhizium anisopliae* Mafz 01 (52.5 ± 6.45%), *B. brongniartii* 400 (52.5 ± 6.45%), and *B. bassiana* Dsxj-07 (73.75 ± 6.29%) at 1.0 × 10^9^ spores/mL [41]. Additionally, *B. bassiana* PfBb has high compatibility with insecticides (e.g., beta cypermethrin), which could increase the efficiency of insecticides [42]. However, the lethal effect of *B. bassiana* PfBb on other lepidopterans has not been tested. In this study, therefore, we tested the pathogenicity of *B. bassiana* PfBb on *S. frugiperda* larvae of different instars and determined the protective enzymes and detoxifying enzymes in the immune resistance of *S. frugiperda* larvae infected with *B. bassiana* PfBb.

## 2. Materials and Methods

### 2.1. Insects

The larvae of *S. frugiperda* were collected from corn fields at Jiaoyi township (119°28′ E, 22°83′ N), Hengzhou City, Guangxi Zhuang Autonomous Region, China. The larvae were individually reared with corn leaves in plastic Petri dishes (9 cm diameter × 1.5 cm height) at 26 ± 1 °C, with 75 ± 10% RH, and a photoperiod of 14:10 h (L:D) until pupation in the artificial climate chambers (LRH-250, Changzhou Putian Instrument Manufacturing Co., Ltd., Changzhou, China). Newly emerged moths were paired in plastic cups (11.5 cm diameter × 15.5 cm height) and fed with 10% honey solution supplied in a small cotton wick. The cups were replaced with a new one after the eggs laid on the cup wall were observed. The cups with eggs were maintained in plastic boxes (24.5 cm diameter × 12.0 cm height) until the hatching of neonate larvae. *S. frugiperda* larvae were reared on leaves of corn seedlings (Meiyu No. 3; Hainan Lvchuan Seed Co., Ltd., Hainan, China) planted in the laboratory, and newly hatched larvae of the fourth generation were used for the experiments.

### 2.2. Entomopathogenic Fungus

The entomopathogenic fungus *B. bassiana* PfBb strain used in this study was isolated in 2018 from naturally infected dead larvae of *P. flammans* in Guangxi. Before the experiment, the strain was cultured on potato dextrose agar (PDA) medium (200 g potato, 20 g glucose, 15 g agar, 1000 mL water, and natural pH) for 15 days at 25 ± 1 °C, with 80 ± 5% RH and a photoperiod of 0:24 h (L:D). After being cultured for 15 days, the fungal conidia were harvested from the surface of the culture medium. After preparing the conidial suspensions, the conidial concentration was determined using a hemocytometer (MC02270113, Shanghai Qiujing Biochemical Reagent & Instrument Co., Ltd., Shanghai, China) and adjusted to 1 × 10^5^, 1 × 10^6^, 1 × 10^7^, and 1 × 10^8^ spores/mL using Tween-80 (0.05%) solution [43].

### 2.3. Virulence of B. bassiana PfBb on S. frugiperda Larvae

Four conidial concentrations (i.e., treatments: 1 × 10^5^, 1 × 10^6^, 1 × 10^7^, and 1 × 10^8^ spores/mL) with Tween-80 (0.05%) as the control (CK) were set up to test the virulence of *B. bassiana* PfBb on six *S. frugiperda* larval instars. There were three replicates for each treatment, with 30 individuals per treatment. For each treatment, the *S. frugiperda* larvae were immersed individually in the solution for 10 s and placed in a Petri dish. The larvae were reared at 26 ± 1 °C, 75 ± 10% RH, and a photoperiod of 14:10 h (L:D) and checked twice (09:00 and 21:00) a day for seven consecutive days. A larva was regarded as dead if it did not respond when touched with a brush. The number of dead larvae was recorded, and the cumulative mortality was calculated as [(treatment group mortality—control group mortality)/(1—control group mortality) × 100%]. The median lethal concentration (LC_50_) of *S. frugiperda* larvae was calculated based on the cumulative mortality of each treatment during the first seven days. The dead larvae were removed into a culture dish sterilized with 75% alcohol to moisten the culture. If mycelia were observed on the larval integument, they were considered cadavers. The number of insect cadavers was recorded to calculate the cadaver rate (number of dead larvae growing mycelium/total number of dead larvae × 100%) [43]. 

### 2.4. Time-Dose-Mortality (TDM) Model of B. bassiana PfBb Affecting S. frugiperda

A time-dose-mortality model of *B. bassiana* PfBb against *S. frugiperda* larvae was established based on data obtained from the virulence test. LC_50_ and LC_90_ were calculated using the model to simulate time and dose-response parameters. The establishment of the TDM model requires a bioassay, including *i* dose (the dose is *i*), day after treatment is *j*, the cumulative mortality probability *p_ij_* (*p_ij_* = 1-exp[-exp(*τ_j_* + *β*log_10_(*d_j_*))]) caused by any dose *d_i_* (*i* = 1, 2, …, *i*) at any time *t_j_* (*j* = 1, 2, …, *j*), where *β* is the slope of dose effect and *τ_j_* is the cumulative time effect parameter from *t*_1_ to *t_j_*. Note that *p_ij_* is a time-dependent variable. However, the above equation cannot directly fit the bioassay data because the binomial variable, *p_ij_*, for modeling does not satisfy the requirement for independence of time. To guarantee that the observed mortality probability was independent of time, the true mortality that occurred at *d_j_* at the interval [*t_j_*_−1_, *t_j_*] was considered. That is, the conditional death probability *q_ij_*, which can be expressed as *q_ij_* = 1-exp[-exp(*γ_j_* + *β*log_10_(*d_j_*))], where, *γ_j_* is the time effect parameter to be estimated in the time interval [*t_j_*_−1_, *t_j_*] [44]. 

### 2.5. Enzymatic Sample Preparation

The *S. frugiperda* larvae of different instars were treated by the immersion method with 1 × 10^8^ spores/mL of *B. bassiana* PfBb and Tween-80 (0.05%) as the treatment group and control group, respectively. The treated *S. frugiperda* larvae were reared the same way as above. During culturing, the treated larvae were sampled randomly at 12, 24, 36, 48, 60, and 72 h after treatment with *B. bassiana* PfBb or Tween-80 (0.05%). The number of samples for each test group at every time point was 50 (1st and 2nd instars), 10 (3rd and 4th instars), and 5 larvae (5th and 6th instars), respectively. The insect samples were homogenized with 0.1 mol/L PBS (sample weight: PBS = 1 g:9 mL) using an electric homogenizer (JXFSTPRP-24L, Shanghai Jingxin Industrial Development Co., Ltd., Shanghai, China). Next, the tissue homogenate was centrifuged (Centrifuge 5424R, Eppendorf, Hamburg, Germany) at 12,000 rpm and 4 °C for 15 min. The supernatant, as the enzyme source, was transferred to new tubes and stored at −20 °C [45]. Each test was repeated three times, and the enzymatic activities associated with the observed larvae were determined.

### 2.6. Measurement of Enzyme Activities in S. frugiperda Larvae

The total protein quantitative assay kit (Product ID: A045-2-2), CAT assay kit (Product ID: A007-1-1), POD assay kit (Product ID: A084-1-1), SOD assay kit (Product ID: A001-1), GSTs assay kit (Product ID: A004-1-1), and CarE assay kit (Product ID: A133-1-1) were purchased from Nanjing Jiancheng Bioengineering Institute, Nanjing, China. The insect CYP450 ELISA Kit (Product ID: ml036261) was obtained from Shanghai MLBIO Biotechnology Co. Ltd., Shanghai, China. The optical density was recorded using an absorbance microplate reader (Spectra Max Plus 384, Molecular Devices, San Jose, CA, USA).

Activity assays of each enzyme were carried out by following the manufacturer’s instructions. The NH_3_^+^ group of the protein molecule can be combined with the anion of Coomassie brilliant blue to make the solution turn blue, and the protein content can be calculated by measuring the absorbance at 595 nm wavelength after 10 min of reaction. CAT activity determination is defined as the amount of 1 μmol H_2_O_2_ decomposed per milligram of tissue protein per second as 1 unit (U). Briefly, H_2_O_2_ was utilized as a substrate, and the absorbance of the reaction was measured at 405 nm wavelength for 1 min. The determination of POD activity was defined as 1 U per mg of tissue protein per minute to catalyze 1 μg of substrate per minute at 37 °C. In short, the absorbance after 30 min of tissue protein reaction was measured at a wavelength of 420 nm. For SOD activity determination, the amount of SOD corresponding to the SOD inhibition rate of 50% per mg of tissue protein in 1 mL of reaction solution was taken as 1 U. In summary, the absorbance of the enzyme solution after reacting at 37 °C for 40 min was measured at a wavelength of 550 nm. GST activity determination. The non-enzymatic reaction was subtracted from the reaction per mg of tissue protein at 37 °C for 1 min, and the GSH concentration in the reaction system was reduced by 1 μmol/L as 1 U. That is, after the enzyme solution was reacted at 37 °C for 10 min, the absorbance was measured at a wavelength of 412 nm. The CarE activity was determined as 1 U per mg of tissue protein at 37 °C with an increase of 1 in catalytic absorbance per minute. Briefly, the difference in absorbance was measured at a wavelength of 450 nm after the enzyme solution reacted for 10 and 190 s [45]. To determine CYP450 activity, a linear regression equation was made with the standard product. The absorbance of the enzyme solution at 450 nm after reacting at 37 °C for 60 min was measured, and the concentration value and enzyme activity of each sample were calculated according to the equation. CAT, POD, SOD, and GST enzyme activities in *S. frugiperda* larvae were calculated based on their protein content (Appendix A).

### 2.7. Statistical Analyses

The normality and homoscedasticity of all data were tested prior to analysis using the Kolmogorov–Smirnov and Levene’s tests, respectively. Data on cadaver rates and enzyme activities were analyzed using a one-way analysis of variance (ANOVA), followed by Tukey’s honestly significant difference (HSD) multiple tests. The regression equation, LT_50_, LT_90_ and their 95% confidence limits were calculated by Probit regression analysis. The effects of *B. bassiana* PfBb on enzyme activities in infected *S. frugiperda* larvae were analyzed by an independent samples *t-*test. The statistical significance level was determined to be alpha ≤0.05. A probit regression was applied to analyze the virulence of *B. bassiana* PfBb to six instars of *S. frugiperda* larvae, and calculate the LC_50_, LC_90_, and their 95% confidence limits. Survival curves were subjected to a log-rank test and graphed using GraphPad Prism 8.0.2 (GraphPad Software Inc., San Diego, CA, USA). The Hosmer–Lemeshow test was performed on the time-dose-mortality model using DPS 7.05, and a t-test was used to analyze the significance of the dose and time-effect parameters of different instar larvae. Data analyses were performed using SPSS 25.0 (IBM Corp., Armonk, NY, USA).

## 3. Results

### 3.1. Effect of B. bassiana PfBb on S. frugiperda Larvae

The *S. frugiperda* larvae of different instars can be infected by *B. bassiana* PfBb. After the dead larvae were wet-cultured, white hyphae grew on both sides of the integument, and then the hyphae wrapped the whole body (Figure 1). 

### 3.2. Effect of B. bassiana PfBb on Mortality of S. frugiperda Larvae

Mortality of *S. frugiperda* larvae on each instar increased gradually with the increase in the spore concentration of *B. bassiana* PfBb (Log-rank test, first instar: $x_{4}^{2}$ = 81.82, *p* < 0.001; second instar: $x_{4}^{2}$ = 88.07, *p* < 0.001; third instar: $x_{4}^{2}$ = 67.64, *p* < 0.001; fourth instar: $x_{4}^{2}$ = 42.84, *p* < 0.001; fifth instar: $x_{4}^{2}$= 39.21, *p* < 0.001; sixth instar: $x_{4}^{2}$ = 15.17, *p* = 0.004) (Figure 2). For example, the mortality rate of the first instar larvae increased from 25.00% at day 1 to 82.33% at day 5 after being treated at 1.0 × 10^8^ spores/mL (Figure 2A). A similar mortality pattern was also detected for the second to fifth instars (Figure 2B–E). After 7 days of treatment, the cumulative mortality of the sixth instar larvae treated with 1.0 × 10^7^ and 1.0 × 10^8^ spores/mL suspensions was 15.35% and 17.02%, respectively, which are significantly higher than those treated with CK (3.33%), 1.0 × 10^5^ (3.42%), and 1.0 × 10^6^ spores/mL (6.67%) (Figure 2F). Obviously, the cumulative mortality gradually reduced with the increase in the larval instar (Figure 2A–F). After 7 days of treatment, the LC_50_ of spore concentrations increased with the instar stage of *S. frugiperda* (Table 1).

### 3.3. Effect of B. bassiana PfBb on the Cadaver Rate of Infected S. frugiperda Larvae

For a given larval instar (i.e., first to fifth instars), the cadaver rate of instar larvae significantly increased with the increase of spore concentration (first instar: *F*_3,8_ = 33.54, *p* < 0.001; second instar: *F*_3,8_= 30.71, *p* < 0.001; third instar: *F*_3,8_ = 24.27, *p* < 0.001; fourth instar: *F*_3,8_ = 9.22, *p* = 0.006; fifth instar: *F*_3,8_ = 14.08, *p* = 0.002) (Figure 3). For example, the cadaver rate of the treatment of 1.0 × 10^8^ spores/mL (76.67 ± 5.77%) was significantly higher than those other treatment concentrations at the first instar larvae (Figure 3). The cadaver rate of the 4th (36.67 ± 7.64%) and 5th (33.33 ± 7.64%) instar larvae under the treatment of 1.0 × 10^8^ spores/mL was also significantly higher than other treatment concentrations (Figure 3). However, there was no significant difference in the cadaver rate of the sixth instar larvae between different spore concentrations (*F*_3,8_ = 3.42, *p* = 0.073) (Figure 3).

### 3.4. Time-Dose-Mortality (TDM) Model of B. bassiana PfBb on S. frugiperda Larvae

The cumulative mortality data of 1–5 days for the first to second instar larvae, 1–6 days for the third to fifth instar larvae, and 2–7 days for the sixth instar larvae were applied to the TDM model for analysis according to the mortality of *S. frugiperda* larvae before and after infection with spores of *B. bassiana* PfBb. The Hosmer–Lemeshow test for the heterogeneity of the goodness of fit for the binomial variable *pij* was not significant (first instar: $X_{7}^{2}$ = 0.45, *p* = 0.999; second instar: $X_{7}^{2}$ = 2.63, *p* = 0.917; third instar: $X_{7}^{2}$ = 1.72, *p* = 0.974; fourth instar: $X_{8}^{2}$ = 1.93, *p* = 0.983; fifth instar: $X_{8}^{2}$= 2.13, *p* = 0.977; sixth instar: $X_{9}^{2}$= 4.09, *p* = 0.905), indicating that the data fit the model well (Table 2). The cumulative effect parameter *τ_j_* increased with the increase in time, indicating that the dose effect, time effect, and their interaction effect significantly influenced the mortality of *S. frugiperda* larvae. The slopes (*β*) of dose response of first to sixth instar larvae to *B. bassiana* PfBb were 0.4068, 0.5825, 0.5602, 0.6144, 0.6952, and 0.4701, respectively, suggesting that the fifth instar larvae were the most sensitive to the increase of *B. bassiana* PfBb spore concentration. The time effect parameters of first and second instar larvae to *B. bassiana* PfBb reached the maximum on the third day after infection (*γ*_3_), and that of third instar larvae on the second day (*γ*_2_), fourth and fifth instar larvae on the fourth day (*γ*_4_), and sixth instar larvae on the fifth day after infection (*γ*_5_).

### 3.5. Effect of B. bassiana PfBb Concentration on the Lethality of S. frugiperda Larvae

According to the TDM model, the estimated dose effect of *S. frugiperda* larvae infected with *B. bassiana* PfBb is shown in Figure 4. With the extension of time, the lethal concentration decreased, and the dose effect increased (Figure 4). With the increase in lethal concentration, the time effect increased (Figure 4). However, the actual maximum cumulative mortality of the first to sixth instar larvae is not up to 90%. Therefore, the estimated LC_90_ lethal concentrations corresponding to the first to sixth instar larvae were rather large (Figure 4). 

### 3.6. Effect of B. bassiana PfBb on CAT Enzyme Activity in S. frugiperda Larvae

The CAT activity in the first to fifth instars of *S. frugiperda* infected by *B. bassiana* PfBb was significantly different over infection time, with a peak at 36, 12, 24, 24, and 60 h, respectively (*F*_5,12_ = 10.76, 10.30, 6.91, 5.17, and 3.32, respectively; *p* < 0.05); however, CAT activity in the sixth instar did not vary over the infection time (*F*_5,12_ = 1.82, *p* = 0.184) (Figure 5). Compared to that in the control, CAT activity was significantly higher at 24 h (*t*_4_ = 7.23, *p* = 0.002) and 36 h (*t*_4_ = 6.53, *p* = 0.003) in the first instar (Figure 5A), at 12 h (*t*_4_ = 4.08, *p* = 0.015), 24 h (*t*_4_ = 4.87, *p* = 0.008) and 36 h (*t*_4_ = 5.10, *p* = 0.007) in the second instar (Figure 5B), at 24 h in the third instar (*t*_4_ = 5.43, *p* = 0.006), and at 36 h (*t*_4_ = 4.00, *p* = 0.016) and 48 h (*t*_4_ = 3.85, *p* = 0.018) in the fourth instar (Figure 5C,D), with a significantly lower CAT activity detected at 24 h in the fifth instar (*t*_4_ = −4.12, *p* = 0.015) (Figure 5E).

### 3.7. Effect of B. bassiana PfBb on POD Enzyme Activity in S. frugiperda Larvae

The POD activity in the first to sixth instar larvae of *S. frugiperda* infected by *B. bassiana* PfBb significantly changed over the infection time, with a peak at 24, 12, 24, 36, 24, and 24 h, respectively (*F*_5,12_ = 13.14, 14.64, 4.06, 7.12, 3.77, and 5.36, respectively; *p* < 0.05) (Figure 6). Compared to the control, POD activity was significantly higher at 24 h (*t*_4_ = 3.42, *p* = 0.027), 36 h (*t*_4_ = 3.85, *p* = 0.018), and 48 h (*t*_4_ = 3.13, *p* = 0.035) in the first instar (Figure 6A), at 12 h (*t*_4_ = 5.32, *p* = 0.006) in the second instar (Figure 6B), at 12 h in the third instar (*t*_4_ = 4.33, *p* = 0.012) (Figure 6C), and at 48 h in the fourth instar (*t*_4_ = 3.97, *p* = 0.017) (Figure 6D); while in the sixth instar, it was significantly lower at 36 h (*t*_4_ = −5.846, *p* = 0.004) and 48 h (*t*_4_ = −3.025, *p* = 0.033) (Figure 6F).

### 3.8. Effect of B. bassiana PfBb on SOD Enzyme Activity in S. frugiperda Larvae

The SOD activity in *S. frugiperda* larvae did not significantly vary over the infection time in the first and fifth instars (*F*_5,12_ = 0.92 and 2.20, respectively; *p* > 0.05) but significantly changed over time in the second, third, fourth, and sixth instars with a peak detected at 48, 24, 24, and 48 h, respectively (*F*_5,12_ = 11.28, 5.16, 4.46, and 4.49, respectively; *p* < 0.05) (Figure 7). Compared to that in the control, SOD activity was significantly higher at 12 h (*t*_4_ = 3.19, *p* = 0.033), 24 h (*t*_4_ = 4.99, *p* = 0.008), and 36 h (*t*_4_ = 4.98, *p* = 0.008) in the first instar (Figure 7A), and at 12 h (*t*_4_ = 3.51, *p* = 0.025), 36 h (*t*_4_ = 3.40, *p* = 0.027), and 48 h (*t*_4_ = 8.39, *p =* 0.001) in the second instar (Figure 7B); while in the fourth instar larvae it was significantly lower at 12 h (*t*_4_ = −4.18, *p* = 0.014) but higher at 60 h (*t*_4_ = 3.579, *p* = 0.023) (Figure 7D), and in the fourth instar it was significantly lower at 12 h (*t*_4_ = −5.94, *p* = 0.004) (Figure 7F).

### 3.9. Effect of B. bassiana PfBb on GST Enzyme Activity in S. frugiperda Larvae

The GST activity in the first to fifth instars infected by *B. bassiana* PfBb significantly changed over infection time, with a peak detected at 36, 12, 12, 36, and 24 h, respectively (*F*_5,12_ = 12.40, 10.04, 8.65, 8.08, and 11.40, respectively; *p* < 0.05), while in the sixth instar it did not significantly vary over infection time (*F*_5,12_ = 2.15, *p* = 0.129) (Figure 8). Compared to that in the control, GST activity was significantly higher at 12 h (*t*_4_ = 4.40, *p* = 0.012), 24 h (*t*_4_ = 6.26, *p* = 0.003), 36 h (*t*_4_ = 11.29, *p* < 0.001), and 60 h (*t*_4_ = 4.32, *p* = 0.013) in the first instar (Figure 8A), at 12 h (*t*_4_ = 6.22, *p* = 0.003), and 24 h (*t*_4_ = 5.20, *p* = 0.007) in the second instar (Figure 8B), at 12 h (*t*_4_ = 4.97, *p* = 0.008) and 36 h (*t*_4_ = 4.97, *p* = 0.008) in the third instar (Figure 8C), at 36 h (*t*_4_ = 3.31, *p* = 0.030) and 48 h in the fourth instar (*t*_4_ = 3.09, *p* = 0.037) (Figure 8D), and at 24 h in the fifth and sixth instars (*t*_4_ = 5.48 and 3.94, respectively; *p* < 0.05) (Figure 8E,F).

### 3.10. Effect of B. bassiana PfBb on CarE Enzyme Activity in S. frugiperda Larvae

The CarE activity in the first four and the sixth instars significantly changed over infection time, with a peak detected at 36, 36, 24, 48, and 60 h, respectively (*F*_5,12_ = 4.22, 4.18, 6.06, 3.73, and 5.95, respectively; *p* < 0.05), while in the fifth instar it did not significantly vary over the infection time (*F*_5,12_ = 1.23, *p* = 0.353) (Figure 9). Compared to that in the control, CarE activity was significantly higher at 36 h (*t*_4_ = 3.40, *p* = 0.027) in the first instar (Figure 9A), and at 12 h in the sixth instar (*t*_4_ = 4.80, *p* = 0.009) (Figure 9F); however, it was significantly lower at 72 h in the second instar (*t*_4_ = −3.24, *p* = 0.032) (Figure 9B), and at 36 h (*t*_4_ = −3.47, *p* = 0.026) and 48 h (*t*_4_ = −4.07, *p* = 0.015) in the third instar (Figure 9C).

### 3.11. Effect of B. bassiana PfBb on CYP450 Enzyme Activity in S. frugiperda Larvae

The CYP450 activity in the first five instars significantly changed over infection time (*F*_5,12_ = 6.87, 4.24, 19.00, 6.91 and 7.83, respectively; *p* < 0.05), while in the sixth instar it did not significantly vary over infection time (*F*_5,12_ = 1.05, *p* = 0.435) (Figure 10). Compared to that in the control, CYP450 activity was significantly lower at 24 h (*t*_4_ = −8.04, *p* = 0.001), 36 h (*t*_4_ = −5.34, *p* = 0.006), and 48 h (*t*_4_ = −8.17, *p* = 0.001) in the first instar (Figure 10A), but significantly higher at 12 h (*t*_4_ = 4.60, *p* = 0.010), 24 h (*t*_4_ = 4.33, *p* = 0.012), and 36 h (*t*_4_ = 3.95, *p* = 0.017) in the second instar (Figure 10B), at 12 h (*t*_4_ = 6.37, *p* = 0.003) and 24 h (*t*_4_ = 8.85, *p* = 0.001) in the third instar (Figure 10C), at 12 h in the fourth instar (*t*_4_ = 6.34, *p* = 0.003) (Figure 10D), and at 12 h (*t*_4_ = 5.36, *p* = 0.006) and 24 h (*t*_4_ = 3.02, *p* = 0.039) in the fifth instar (Figure 10E).

## 4. Discussion

*Beauveria bassiana* is a promising biological control agent for insect pests. In this study, we show that *B. bassiana* PfBb, a strain collected from *P. flammans* larvae, had strong pathogenicity to the young larvae of *S. frugiperda*, and that the protective enzymes and detoxifying enzymes in *S. frugiperda* larvae play important roles in resisting the infection of *B. bassiana* PfBb, especially on young larvae. Our results show that the pathogenicity of *B. bassiana* PfBb on non-target host *S. frugiperda,* as well as the protective enzymes and detoxifying enzymes in immune resistance, are instar-stage dependent.

Virulence is a major measure of pathogenicity in entomopathogenic fungi. The degree of virulence is directly related to the ability of entomopathogenic fungi to reduce insect mortality despite the existence of host resistance. The results of the current study show that *B. bassiana* PfBb, a strain collected from *P. flammans* larvae, had strong virulence to first to third instar larvae of *S. frugiperda,* though its pathogenicity gradually decreased with the increase of the larval instars (Figure 2 and Figure 3; Table 1 and Table 2). Similar results have also been reported when pathogenic fungi are collected from other host species [35] or when other pest insects are infected with *B. bassiana* [46,47]. For example, the application of *B. bassiana* has a higher effect on the young larvae of *Indarbela dea* [48], suggesting younger larvae are more susceptible to *B. bassiana* than older larvae [49]. This may be attributed to four reasons. First, the insect epidermis is the first barrier to pathogenic fungal infection [50]. The pathogenicity of entomogenous fungi to pests often decreases with the increase of larval instars due to the higher content of melanin in the insect epidermis and midgut of older instars, hindering fungal budding [51]. Second, the virulence of entomogenous fungi to pests is also closely related to the structure of the larval body wall [52], because the body wall of young larvae is relatively thin, while the waxy layer of the body wall gradually thickens with the increase of the instar stage, which prevents the invasion of *B. bassiana* [53]. Third, the developmental period of the first to sixth instar larvae of *S. frugiperda* is 3.0, 2.0, 2.0, 1.1, 1.5, and 4.3 d, respectively, at 25 °C [54]. We speculate that the time period of spores attaching to the body surface of older instars is shorter due to the shortening molting process. Fourth, insects can overcome different poisons by regulating changes in protective and detoxifying enzymes in the body [55,56]. The reduced pathogenicity of *B. bassiana* PfBb to older *S. frugiperda* larvae in this study may be due to the enzyme systems in the non-target host (see discussion below).

After the pathogen successfully invades a host, it activates the host’s defense system, resulting in changes in the host’s protective enzymes, especially SOD, POD, and CAT. It is well known that a large quantity of active oxygen will accumulate in the insect bodies, which stimulates an antioxidant enzyme response when insects suffer from stress caused by adverse factors [19,57]. Earlier studies have shown that SOD and CAT activities are inversely correlated with larval instars [58]. In this study, our results illustrate that the activity of protective enzymes and reaction times in different larval instars varied differently after being infected with *B. bassiana* PfBb. These findings indicate that young larvae are more sensitive faster response time to *B. bassiana* PfBb. The protective enzyme activity in the first and second larval instars significantly increased first and then decreased. Similar results have been reported in *Bemisia tabaci*, *Mythimna separata*, and *Xylotrechus rusticus* [19,59,60]. For example, *B. bassiana* leads to an increase in reactive oxygen species in *X. rusticus* larvae in the early stage of infection and a quick activation of the antioxidant enzyme system [19]. However, the infection of pathogenic fungi leads to the inhabitation of antioxidant enzyme activity in the late instar stages of *S. frugiperda* (Figure 5, Figure 6 and Figure 7), so the ability of infected larvae to scavenging free radicals in the body is weakened. Previous studies have shown that the oxygen balance in the body is easily disrupted when insects are infected with *B. bassiana* [21,61]. For example, SOD converts the superoxide radical O_2_ into H_2_O_2_ when insects are stimulated by exogenous compounds; however, CAT decomposes H_2_O_2_ into H_2_O and O_2_ when the concentration of H_2_O_2_ is too high, while POD decomposes when the concentration of H_2_O_2_ is low [62,63]. Moreover, the activity of protective enzymes determines the intensity of external stimuli [23], insecticide resistance, and stress resistance of insects [58,64]. In this study, we revealed that SOD and CAT activity in the fifth and sixth instars were not significantly different compared with the control. We speculate that *B. bassiana* PfBb has weaker stimulation to the older stage of non-target hosts and lower H_2_O_2_ concentration after infection, resulting in insignificant changes in the activity of protective enzymes. However, we only measured the activity of protective enzymes in susceptible larvae of *S. frugiperda*, and the changes in oxygen free radicals and gene expression in susceptible larvae need further research.

To inhibit further damage by pathogenic fungi, detoxification enzymes, such as CarE, GST, and CYP450, in insects can efficiently metabolize exogenous toxic compounds [17]. Theoretically, insects resist the poisoning of different exogenous substances by enhancing the activity of detoxification enzymes and promoting the expression of detoxification enzyme genes, thereby improving their resistance to pathogenic fungi. In the present study, the detoxification enzyme activity and reaction time in different larval instars varied differently after being infected with *B. bassiana* PfBb. The detoxification enzyme activity in the first to fourth larval instars changed significantly compared with the control (Figure 8, Figure 9 and Figure 10). Similar findings have also been found in *X. rusticus* and *B. tabaci* [19,59]. Our results revealed that young larvae are more susceptible and have a faster reaction time to *B. bassiana*. GST is an important detoxification enzyme in the metabolism of endogenous and exogenous substances in insects, and the increase in GST activity can be used as a sensitive indicator of tissue damage [65]. Our results illustrate that GST in *S. frugiperda* larvae was activated after *B. bassiana* PfBb was inoculated, but because the tissue was destroyed by poison, the GST synthesis capability was reduced with infection time, resulting in the GST increasing first and then decreasing (Figure 8). CarE is a specific catalyzing ester bond hydrolase that not only participates in lipid metabolism but also acts as a detoxification enzyme to metabolize exogenous toxins [16]. Our results show that CarE activity increased first and then decreased with increasing infection time (Figure 9), which is consistent with a previous report [19]. In contrast, the activity of CarE first decreased and then increased after being infected by *B. bassiana,* which is also reported in *P. flammans* [66]. The opposed CarE activity in target and non-target hosts could be attributed to their different resistances to the *B. bassiana* PfBb strain. Empirical evidence demonstrates that numerous responsive genes are differentially expressed in various hosts during infection by *B. bassiana*. For example, the increased or decreased expression of some special genes [i.e., the adhesion (Mad1), protease (Pr2), and secretory lipase] is an important factor for the difference in pathogenicity of *B. bassiana* to different hosts [13]. The differential expression of relative genes in target and non-target hosts infected by the *B. bassiana* PfBb strain needs further study. CYP450 is also a key enzyme involved in detoxification metabolism in insects. In this study, the CYP450 enzyme activity of the first instar larvae in the treatment was significantly lower than that of the control larvae (Figure 10). Cytochrome P450 enzymes are known to activate ecdysone [67]. We suggest that CYP450 enzyme activity decreases after the first instar larvae are infected, and that the synthesis of ecdysone is inhibited, eventually leading to death. Furthermore, the CYP450 enzyme activity in the sixth instar larvae was not significantly different from that in the control (Figure 9), probably due to the fact that the *B. bassiana* PfBb is not activate in the fatty acid degradation pathway, because the differences in cytochrome P450 gene expression are caused by differential degradation of thrips fatty acids of strains with similar virulence [68]. In addition, a study found that the detoxification enzyme activity of *Culex quinquefasciatus* increased with increases in *B. bassiana* concentrations [69]. Therefore, we assume that the pathogenicity of *B. bassiana* PfBb is related to the activity of detoxification enzymes.

## 5. Conclusions

Our findings confirm that the *B. bassiana* PfBb strain can infect all larval instars of *S. frugiperda* and result in a higher cadaver rate, especially for younger larvae. Furthermore, the virulence of *B. bassiana* PfBb to *S. frugiperda* larvae gradually increased with the increase in spore concentration, and the LC_50_ and LC_90_ of *B. bassiana* PfBb for each *S. frugiperda* instar decreased with infection time, indicating a significant dose effect. Although protective enzymes and detoxification enzymes in *S. frugiperda* larvae are usually activated between 12 and 48 h after treatment, the activity of these enzymes in the first three larval instars changed significantly over infection time, while such change in the fifth and sixth larval instars is not obvious. Our results indicate that the pathogenicity of *B. bassiana* PfBb on a non-target host, *S. frugiperda*, is significant but instar-stage dependent. In summary, our results suggest that *B. bassiana* PfBb can be used as a bio-insecticide to control young larvae of *S. frugiperda* in the integrated pest management program.

## Figures and Tables

**Figure 1 insects-13-00914-f001:**
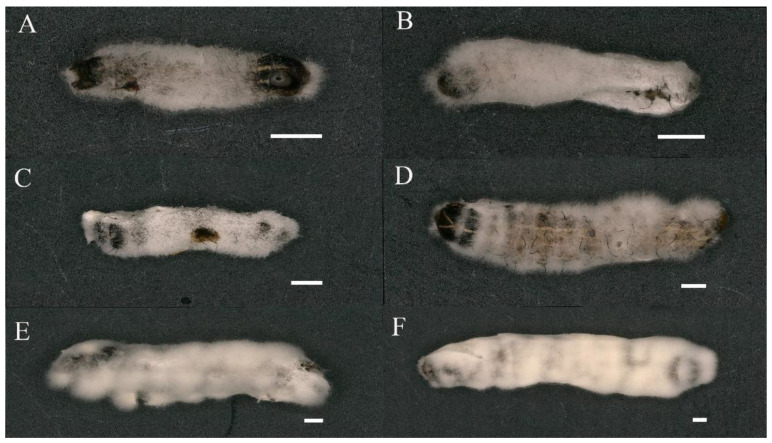
*Spodoptera frugiperda* larvae infected with *Beauveria bassiana* PfBb. (**A**–**F**) represent the first to sixth larval instars, respectively. Scales for different instars are 1 mm.

**Figure 2 insects-13-00914-f002:**
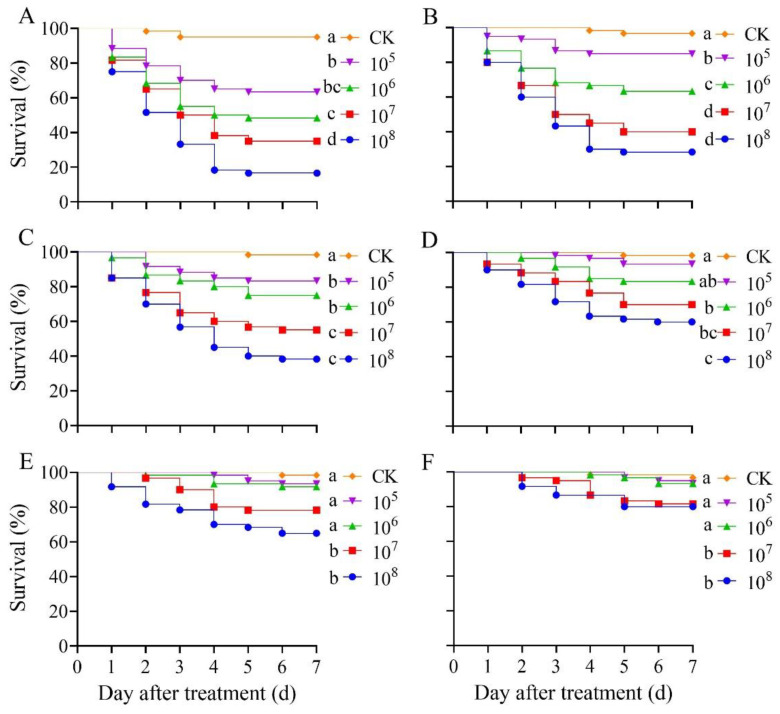
Survival curves of *Spodoptera frugiperda* larvae treated with spore suspensions of different concentrations of *Beauveria bassiana* PfBb. (**A**–**F**) represent the first to sixth larval instars, respectively.

**Figure 3 insects-13-00914-f003:**
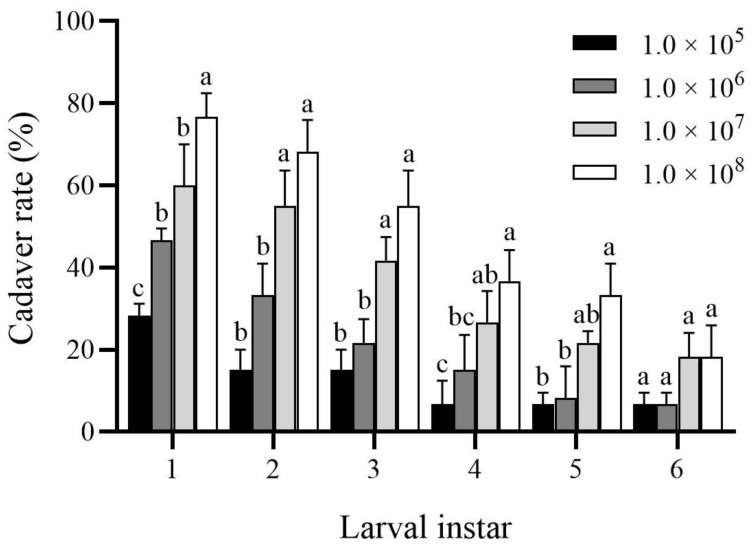
Cadaver rate of *Spodoptera frugiperda* larvae inoculated with different concentrations (spores/mL) of *Beauveria bassiana* PfBb. For each instar, columns with the same letters are not significantly different (Tukey’s HSD test: *p* > 0.05).

**Figure 4 insects-13-00914-f004:**
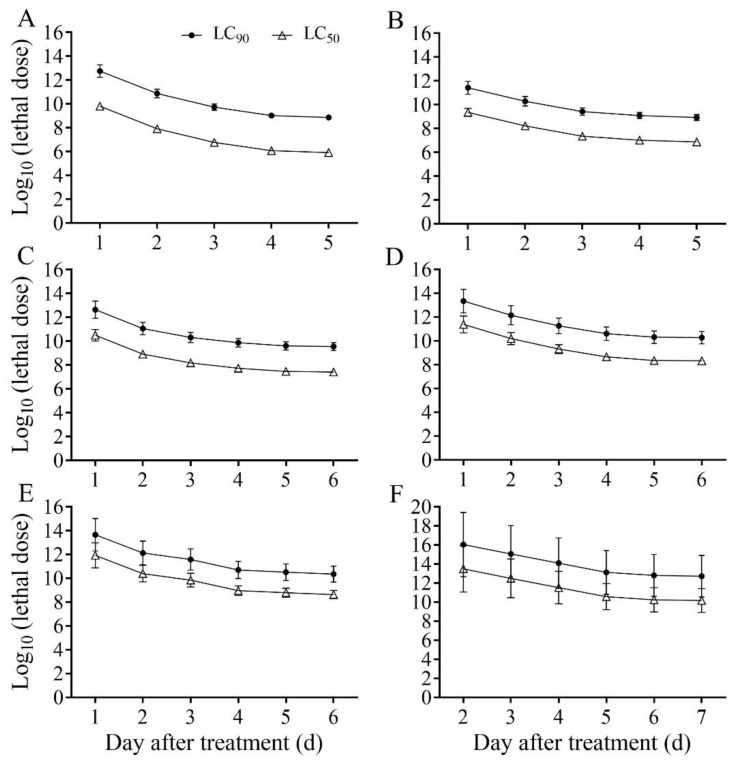
Log_10_ (LC_50_) and Log_10_ (LC_90_) of *Spodoptera frugiperda* larvae infected with *Beauveria bassiana* PfBb over days after treatment. (**A**–**F**) represents the first to sixth instar larvae, respectively.

**Figure 5 insects-13-00914-f005:**
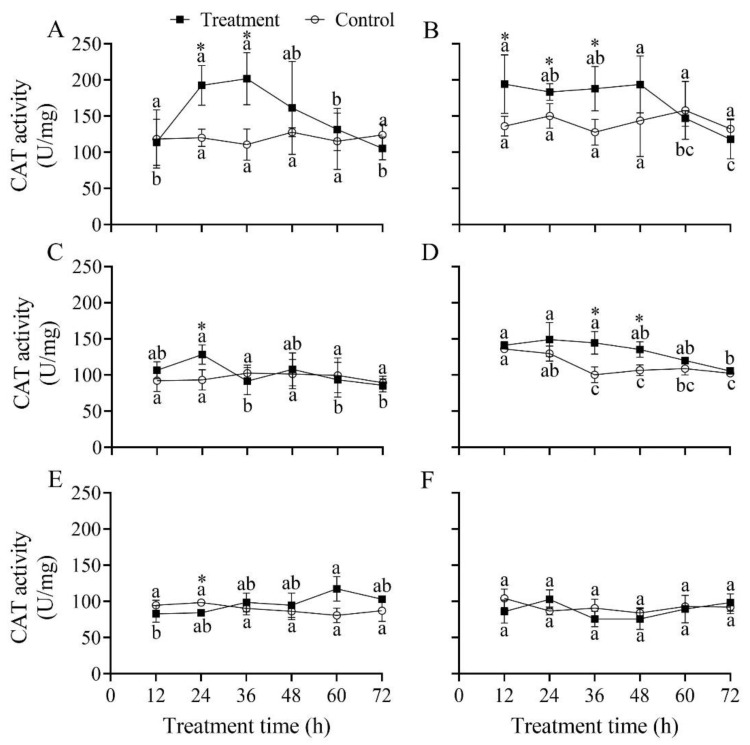
Effect of *Beauveria bassiana* PfBb on CAT activity in *Spodoptera frugiperda*. (**A**–**F**) represent the first to sixth larval instars, respectively. Means (± SD), followed by different lowercase letters in the control or treatment line, are significantly different (Tukey’s HSD test: *p* < 0.05). Asterisks indicate a significant difference between control and treatment at a given time (*t*-test: *p* < 0.05).

**Figure 6 insects-13-00914-f006:**
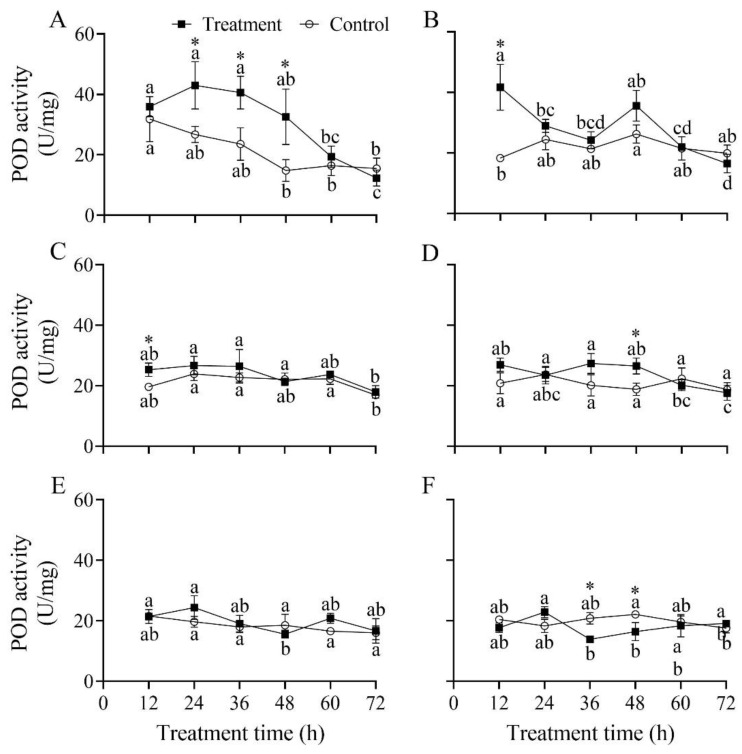
Effect of *Beauveria bassiana* PfBb on POD activity in *Spodoptera frugiperda*. (**A**–**F**) represents the first to sixth instar larvae, respectively. Means (± SD), followed by different lowercase letters in the control or treatment line, are significantly different (Tukey’s HSD test: *p* < 0.05). Asterisks indicate a significant difference between control and treatment at a given time (*t*-test: *p* < 0.05).

**Figure 7 insects-13-00914-f007:**
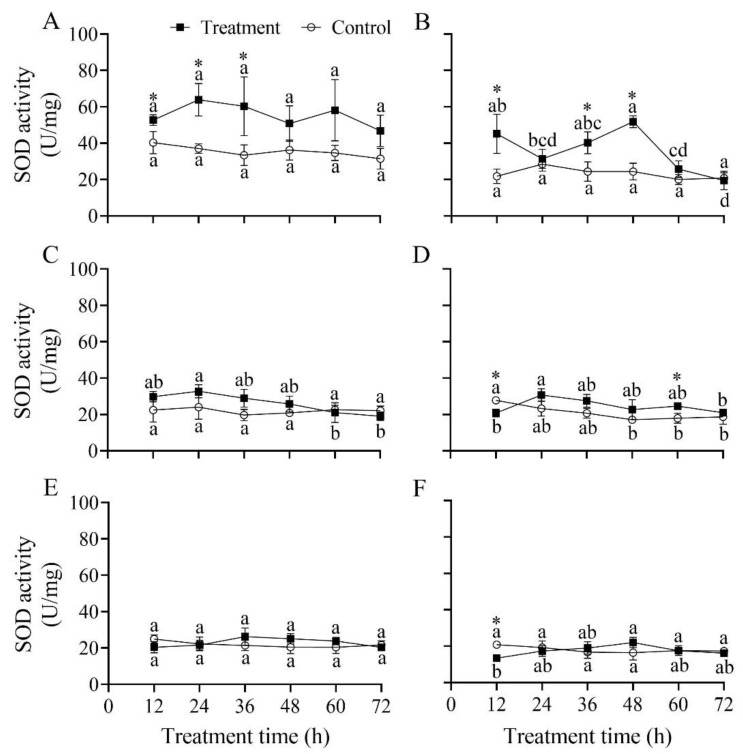
Effect of *Beauveria bassiana* PfBb on SOD activity in *Spodoptera frugiperda*. (**A**–**F**) represents the first to sixth instar larvae, respectively. Means (± SD), followed by different lowercase letters in the control or treatment line, are significantly different (Tukey’s HSD test: *p* < 0.05). Asterisks indicate a significant difference between control and treatment at a given time (*t*-test: *p* < 0.05).

**Figure 8 insects-13-00914-f008:**
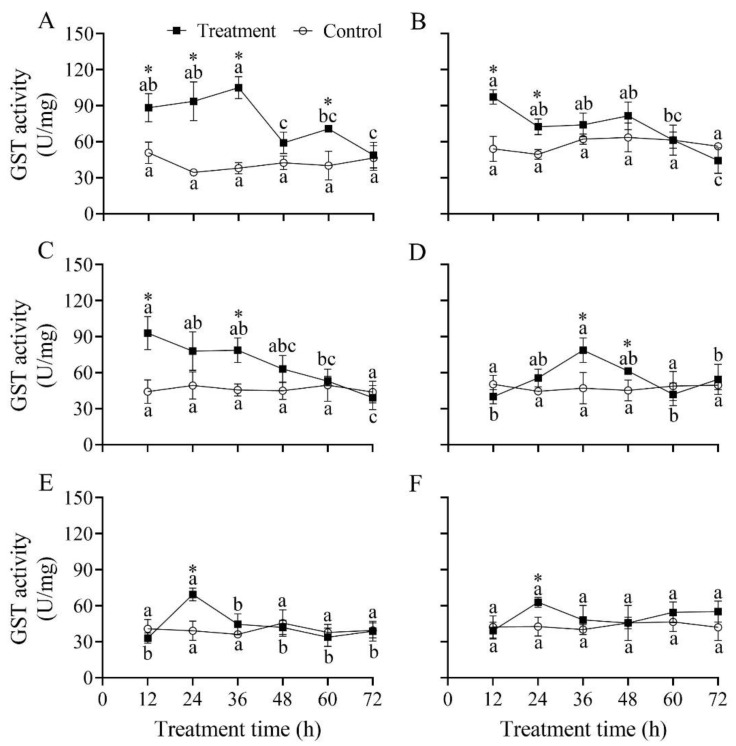
Effect of *Beauveria bassiana* PfBb on GST activity in *Spodoptera frugiperda*. (**A**–**F**) represents the first to sixth instar larvae, respectively. Means (± SD), followed by different lowercase letters in the control or treatment line, are significantly different (Tukey’s HSD test: *p* < 0.05). Asterisks indicate a significant difference between control and treatment at a given time (*t*-test: *p* < 0.05).

**Figure 9 insects-13-00914-f009:**
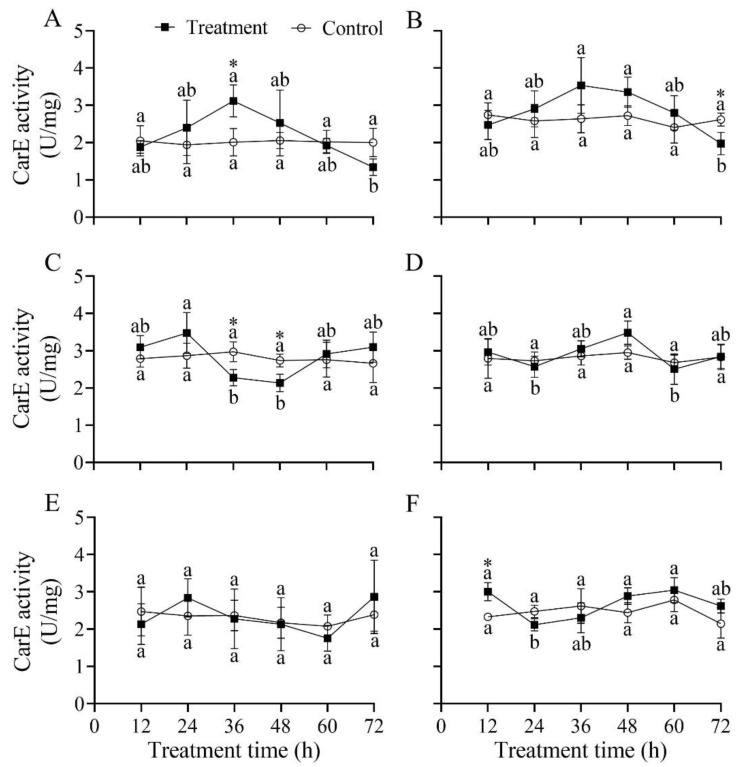
Effect of *Beauveria bassiana* PfBb on CarE activity in *Spodoptera frugiperda*. (**A**–**F**) represents the first to sixth instar larvae, respectively. Means (± SD), followed by different lowercase letters in the control or treatment line, are significantly different (Tukey’s HSD test: *p* < 0.05). Asterisks indicate a significant difference between control and treatment at a given time (*t*-test: *p* < 0.05).

**Figure 10 insects-13-00914-f010:**
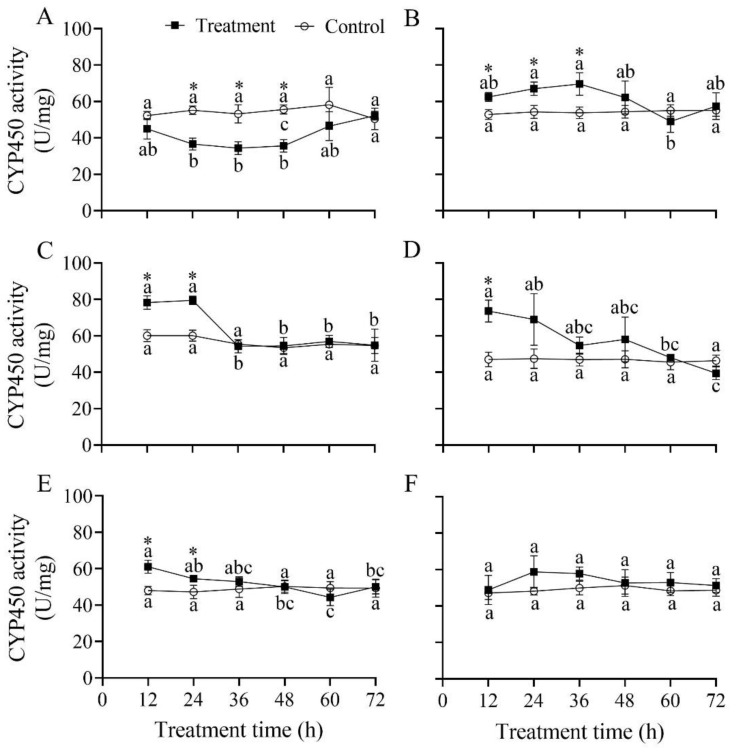
Effect of *Beauveria bassiana* PfBb on CYP450 activity in *Spodoptera frugiperda*. (**A**–**F**) represents the first to sixth instar larvae, respectively. Means (± SD), followed by different lowercase letters in the control or treatment line, are significantly different (Tukey’s HSD test: *p* < 0.05). Asterisks indicate a significant difference between control and treatment at a given time (*t*-test: *p* < 0.05).

**Table 1 insects-13-00914-t001:** Toxicity of *Beauveria bassiana* PfBb to *Spodoptera frugiperda* larvae.

Instar	Regression Equation	*χ* ^2^	*p*	LC_50_ (×10^5^ Spores/mL)	95% Confidence Limit (Spores/mL)
First	y = 0.42*x* − 2.48	0.46	0.796	7.68	2.13 × 10^5^~1.92 × 10^6^
Second	y = 0.54*x* − 3.66	1.23	0.541	54.64	2.62 × 10^6^~1.23 × 10^7^
Third	y = 0.44*x* − 3.22	0.36	0.836	220.11	8.75 × 10^6^~8.88 × 10^7^
Fourth	y = 0.40*x* − 3.43	0.50	0.781	3084.06	7.08 × 10^7^~7.69 × 10^9^
Fifth	y = 0.41*x* − 3.67	0.78	0.678	9579.65	1.62 × 10^8^~6.91 × 10^10^
Sixth	y = 0.23*x* − 2.62	0.50	0.780	2548351.52	1.66 × 10^9^~1.49 × 10^31^

y is the expected mortality and *x* is the logarithm of concentration. All data were calculated using the mortality rates on the seventh day after treatment.

**Table 2 insects-13-00914-t002:** Parameters estimated by fitting the time-does-mortality (TDM) model to assay data of *Beauveria bassiana* PfBb on *Spodoptera frugiperda* larvae.

Instar	Conditional Mortality Model					Cumulative Mortality Model			
	Parameter	Value	SE	*t*	*P*	Parameter	Value	Var (*τ*)	Cov (*β*, *τ*)
First	*β*	0.41	0.14	2.99	0.010	*β*	0.41	0.00	0.00
	*γ* _1_	−4.35	0.98	4.46	0.001	*τ* _1_	−4.35	0.06	−0.01
	*γ* _2_	−4.21	0.97	4.32	0.001	*τ* _2_	−3.59	0.06	−0.01
	*γ* _3_	−4.11	0.97	4.23	0.001	*τ* _3_	−3.12	0.06	−0.01
	*γ* _4_	−4.24	0.99	4.28	0.001	*τ* _4_	−2.84	0.06	−0.01
	*γ* _5_	−5.55	1.18	4.70	0.000	*τ* _5_	−2.77	0.06	−0.01
Second	*β*	0.58	0.17	3.53	0.003	*β*	0.58	0.00	0.00
	*γ* _1_	−5.82	1.20	4.83	0.000	*τ* _1_	−5.82	0.25	−0.03
	*γ* _2_	−5.88	1.22	4.82	0.000	*τ* _2_	−5.15	0.24	−0.03
	*γ* _3_	−5.59	1.19	4.69	0.000	*τ* _3_	−4.65	0.24	−0.03
	*γ* _4_	−6.18	1.26	4.91	0.000	*τ* _4_	−4.46	0.23	−0.03
	*γ* _5_	−6.84	1.34	5.10	0.000	*τ* _5_	−4.37	0.23	−0.03
Third	*β*	0.56	0.18	3.07	0.007	*β*	0.56	0.00	0.00
	*γ* _1_	−6.25	1.38	4.54	0.000	*τ* _1_	−6.25	0.27	−0.03
	*γ* _2_	−5.89	1.32	4.47	0.000	*τ* _2_	−5.36	0.25	−0.03
	*γ* _3_	−6.03	1.35	4.47	0.000	*τ* _3_	−4.95	0.24	−0.03
	*γ* _4_	−6.19	1.36	4.56	0.000	*τ* _4_	−4.69	0.24	−0.03
	*γ* _5_	−6.53	1.38	4.74	0.000	*τ* _5_	−4.55	0.24	−0.03
	*γ* _6_	−8.00	1.83	4.38	0.000	*τ* _6_	−4.52	0.24	−0.03
Fourth	*β*	0.61	0.25	2.50	0.023	*β*	0.61	0.01	0.01
	*γ* _1_	−7.36	1.90	3.88	0.001	*τ* _1_	−7.36	0.55	−0.07
	*γ* _2_	−7.29	1.86	3.92	0.001	*τ* _2_	−6.63	0.51	−0.07
	*γ* _3_	−6.95	1.81	3.85	0.001	*τ* _3_	−6.09	0.49	−0.07
	*γ* _4_	−6.79	1.79	3.80	0.001	*τ* _4_	−5.68	0.48	−0.07
	*γ* _5_	−7.28	1.80	4.05	0.001	*τ* _5_	−5.50	0.47	−0.07
	*γ* _6_	−9.34	2.61	3.58	0.002	*τ* _6_	−5.48	0.47	−0.07
Fifth	*β*	0.70	0.29	2.39	0.029	*β*	0.70	0.02	0.02
	*γ* _1_	−8.65	2.35	3.68	0.002	*τ* _1_	−8.65	1.32	−0.15
	*γ* _2_	−8.01	2.22	3.61	0.002	*τ* _2_	−7.58	1.17	−0.15
	*γ* _3_	−8.36	2.24	3.73	0.002	*τ* _3_	−7.21	1.14	−0.15
	*γ* _4_	−7.37	2.12	3.49	0.003	*τ* _4_	−6.59	1.09	−0.15
	*γ* _5_	−8.64	2.21	3.91	0.001	*τ* _5_	−6.47	1.07	−0.15
	*γ* _6_	−8.60	2.23	3.86	0.001	*τ* _6_	−6.36	1.07	−0.15
Sixth	*β*	0.47	0.33	1.42	0.173	*β*	0.47	0.03	0.03
	*γ* _2_	−6.70	2.52	2.66	0.016	*τ* _2_	−6.70	1.83	−0.23
	*γ* _3_	−7.23	2.58	2.80	0.012	*τ* _3_	−6.23	1.78	−0.23
	*γ* _4_	−6.79	2.40	2.83	0.012	*τ* _4_	−5.78	1.68	−0.22
	*γ* _5_	−6.34	2.37	2.67	0.016	*τ* _5_	−5.33	1.63	−0.22
	*γ* _6_	−7.12	2.35	3.03	0.008	*τ* _6_	−5.18	1.59	−0.22
	*γ* _7_	−8.51	2.90	2.94	0.009	*τ* _7_	−5.14	1.58	−0.22

The subscript number of parameter symbol indicates the days after being treated.

## Data Availability

The data that are presented in this study are available in the article.

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
