# Peer review of "Pathogenicity of Beauveria bassiana PfBb and Immune Responses of a Non-Target Host, Spodoptera frugiperda (Lepidoptera: Noctuidae)"

_insects, 2022, doi:10.3390/insects13100914_

Round 1

Reviewer 1 Report

I am happy to see the elaboration of the manuscript. But, it has some scientific and grammatical issues in the current version, and I have highlighted some issues here. Therefore, the present draft needs minor revision before further process.

Specific comments:

Line. 119-120. Please rephrase it, and clarify it.

Line 129. Hemocytometer, put all data (model, city, country)

Please. Change “hours” by “h”

Please. Change “minutes” by “min”

Sec. 2.2. Put some reasonable reference

Sec. 2.3.; 2.4.; 2.5.; and 2.6.  Put some reasonable references

Line 148-151. Please adjust in statistical art

Conclusion. Please put more explanations

Moreover, the following references could be cited;

https://doi.org/10.1016/j.micpath.2018.04.019

https://doi.org/10.1016/j.cbpc.2021.109112

Author Response

Q1. Line. 119-120. Please rephrase it, and clarify it.
Response: Done (Lines 119-123).

Q2. Line 129. Hemocytometer, put all data (model, city, country).
Response: Done (Lines 133).

Q3. Please. Change “hours” by “h”. Please. Change “minutes” by “min”.
Response: Done.

Q4. Sec. 2.2. Put some reasonable reference. Sec. 2.3.; 2.4.; 2.5.; and 2.6. Put some reasonable references.
Response: Thanks for your suggestions. We have added two references [43] and [45] in the text according to your comments.

Q5. Line 148-151. Please adjust in statistical art.
Response: Done (Lines 228-232).

Q6. Conclusion. Please put more explanations.
Response: We have provided more explanations for the results from the current study (Lines 510-513).

Q7. Moreover, the following references could be cited.
https://doi.org/10.1016/j.micpath.2018.04.019
https://doi.org/10.1016/j.cbpc.2021.109112
Response: We have cited these two papers in the text, i.e., [43] and [45].

Reviewer 2 Report

Some changes should be brought :

- same number of decimals in results in tables,

- only 3 decimals for P values though the paper,

- full latin names are needed in table and figure legends.

Author Response

Q1. Same number of decimals in results in tables.

Response: In the revised version, two decimals were remained in the text and tables .

Q2. Only 3 decimals for P values though the paper.

Response: Done.

Q3. Full latin names are needed in table and figure legends.

Response: Done.